# SnapQuant: A Probabilistic and Nested Parameterization for Binary Networks

## Abstract

In this paper, we study the problem of training real binary weight networks (without layer-wise or filter-wise scaling factors) from scratch under the Bayesian deep learning perspective, meaning that the final objective is to approximate the posterior distribution of binary weights rather than reach a point estimation. The proposed method, named as **SnapQuant**, has two intriguing features: (1) The posterior distribution is parameterized as a policy network trained with a reinforcement learning scheme. During the training phase, we generate binary weights on-the-fly since what we actually maintain is the policy network, and all the binary weights are used in a burn-after-reading style. At the testing phase, we can sample binary weight instances for a given recognition architecture from the learnt policy network. (2) The policy network, which has a nested parameter structure consisting of layer-wise, filter-wise and kernel-wise parameter sharing designs, is applicable to any neural network architecture. Such a nested parameterization explicitly and hierarchically models the joint posterior distribution of binary weights. The performance of SnapQuant is evaluated with several visual recognition tasks including ImageNet. The code will be made publicly available.

## 1 Introduction

Deep Neural Networks (DNNs) have achieved tremendous success in computer vision Krizhevsky et al. (2012), natural language processing Collobert et al. (2011) and speech recognition Hinton et al. (2012). Regardless of the availability of large-scale labeled data and powerful computational resources, the leading accuracies of DNNs are attributed to the huge number of learnable parameters, ranging from a couple of millions to even hundreds of millions. However, this also brings heavy consumption of memory, compute and power resources, which prohibits their use on the resource-constrained platforms.

Binary Weight Network (BWN) is among the most promising way to ease the deployment of DNNs thanks to its advantages of much smaller storage space and cheaper bit-wise operations over the full-precision (32-bit floating-point) counterpart. Because of this, substantial research efforts have been invested on how to train BWNs either from the pre-trained full-precision models or from scratch. Existing solutions can be grouped into two basic families. The first solution family, such as Soudry et al. (2014), BinaryConnect Courbariaux et al. (2015) and Binarized Neural Networks (BNNs) Courbariaux et al. (2016), strictly forces the weight values to be either $+1$ or $-1$ from the perspective of approximating the posterior distribution of the weights. The second solution family, such as XNOR-Net Rastegari et al. (2016), Zhou et al. (2018) and Wan et al. (2018), directly considers the network binarization problem as a layer-wise approximation of the floating-point weight set by a binarized version (that is called a point estimation in this paper). To address notable accuracy degradation, they add filter-wise or layer-wise scaling factors over binarized weights, introducing additional multiplication operations. Our proposed method belongs to the first solution family, but we strive to explore the problem of training BWNs from scratch, by an association of Reinforcement Learning (RL) J William (1992) and Bayesian deep learning for approximating the posterior distribution of binary weights. Although there are some recent research attempts Lin et al. (2017); Ashok et al. (2017); He & Han (2018) that also use RL to address DNN compression, they focus on how to adapt policy network designs to reduce the complexity of pre-trained network models, mostly from the perspective of weights pruning. Our main contributions are summarized as follows:

- We propose **SnapQuant**, a reinforcement learning method for training BWNs from scratch under the Bayesian deep learning perspective, which approximates the posterior distribution of binary weights instead of a single point estimation. During the training phase, it generates binary weights in a burn-after-reading style by maintaining a policy network that parameterizes this posterior distribution. At the testing phase, it enables to sample binary weight instances for a given recognition architecture from the learnt policy network.

- We propose a **nested parameter structure** for the policy network, which explicitly and hierarchically models the joint posterior distribution of binary weights. Such a nested parameter structure has layer-wise, filter-wise and kernel-wise parameter sharing designs, thus it is applicable to any DNN architecture.

- Experiments conducted on the widely used image classification datasets including ImageNet show that SnapQuant has better performance in comparison to related probabilistic methods.

## 2 RELATED WORKS

Here, we briefly summarize recent advancements in the related field.

**Neural Network Quantization**. Prevalent deep neural networks are usually trained with 32-bit floating-point weights, thus the reduction of weights precision is a natural way to compress and accelerate DNNs. Gong et al. (2014) propose to replace the weights in each fully connected layer of a pre-trained DNN model by a small number of the clustered centroid values obtained from vector quantization techniques. Chen et al. (2015) use hash function to perform weight mapping over fully connected layers. Vanhoucke et al. (2011) adopt 16-bit fixed-point implementation of DNNs, and Gupta et al. (2015) utilize 8-bit fixed-point implementation. To achieve significant reduction of network complexity, there have numerous approaches proposed to train binary or ternary DNNs either from the pre-trained full-precision models or from scratch, including but not limited to Soudry et al. (2014); Courbariaux et al. (2015; 2016); Rastegari et al. (2016); Li & Liu (2016); Zhou et al. (2016); Hubara et al. (2016); Zhu et al. (2017); Zhou et al. (2017); Li et al. (2017a); Zhou et al. (2018); Zhang et al. (2018); Wan et al. (2018). During the training phase, most of these methods maintain full-precision weights and use them for gradients accumulation and weights quantization.

**Reinforcement Learning**. Recently, reinforcement learning J William (1992); Volodymyr et al. (2015) has been used to reduce the complexity of DNNs. Veniat & Denoyer (2017) use RL to learn efficient DNN architecture with a budgeted objective function. Lin et al. (2017) propose to train a policy Recurrent Neural Network (RNN) which can dynamically prune DNNs according to the input data and the corresponding feature maps during the inference phase. Ashok et al. (2017) adopt two policy RNNs for learning reduced network architectures from the larger teachers in an incremental way. He & Han (2018) utilize deep deterministic policy gradient agent to address DNN pruning. These RL based methods share a common feature: they mainly focus on how to adapt policy network designs for compressing pre-trained network models, usually from the perspective of weights pruning.

**Other Methods**. Besides aforementioned approaches, there are also many other methods to improve the efficiency of DNNs. Network pruning is a promising way to transform dense DNN models into sparse versions without loss of predication accuracy. This line of research mainly includes network parameter pruning Yann et al. (1990); Hassibi & Stork (1993); Han et al. (2015); Guo et al. (2016), filter pruning Li et al. (2017b) and channel pruning Hu et al. (2016); He et al. (2017). Knowledge distillation presents another way which allows training an efficient yet accurate student network distilled by the knowledge of a larger pre-trained teacher model Hinton et al. (2014); Romero et al. (2015); Mishra & Marr (2018). For more neural network compression methods, we refer the reader to Sze et al. (2017) and Cheng et al. (2018) for comprehensive reviews.

## 3 PROPOSED METHOD: SNAPQUANT

### 3.1 OVERVIEW

Here we consider the problem of training a binary neural network $f$ for a supervised learning task $f : X \rightarrow Y$, in which $X$ is the training set of the input data points and $Y$ is the corresponding objective. In this study, we focus on supervised image classification tasks, thus $X$ represents

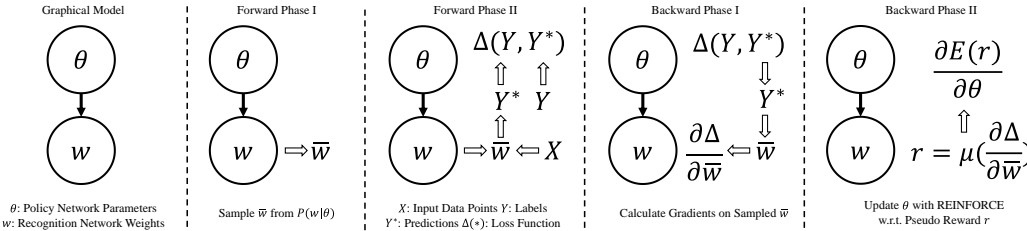

Figure 1: The leftmost panel illustrates the connection between probabilistic nodes. The remaining four panels show how a forward-backward iteration for the likelihood term $\Delta$ is done.

natural images and $Y$ is the set of image labels taken from a pre-defined category taxonomy. Since Convolutional Neural Networks (CNNs) are the most prevalent model, we consider the setting of approximating $f : X \to Y$ with a CNN. More specifically, we restrict the weights of the recognition network (denoted as $w$) to take either value $+1$ or $-1$. We adopt the Bayesian deep learning perspective, in which the ultimate goal is to approximate the posterior distribution $P(w|X, Y)$. Note that most existing methods Rastegari et al. (2016); Li & Liu (2016); Zhou et al. (2016); Hubara et al. (2016); Zhu et al. (2017); Zhou et al. (2017); Li et al. (2017a); Zhou et al. (2018); Zhang et al. (2018); Wan et al. (2018) only give a point estimate of this distribution. The prior distribution of binary weights is denoted as $P(w)$. The likelihood on the training set $\{X, Y\}$ is denoted as $P(Y|w, X)$. According to the Bayesian rule, the concerned posterior can be expressed as $P(w|X, Y) = \frac{P(Y|w,X)P(w)}{P(Y|X)}$. As evaluating the generic posterior $P(w|X, Y)$ is intractable, we resort to the variational approximation $P_\theta(w)$. Though there are other possible probabilistic parameterizations for $\theta$, we propose an alternative that directly parameterizes $P(w|X, Y)$ as a policy network. More specifically, let's denote the parameters of the policy network as $\theta$. As illustrated by the first panel of Fig 1, $w$ is conditioned on $\theta$, or say, $P(w|X, Y) \doteq P_\theta(w)$.

Following the common practice in modern variational approximation Gal & Ghahramani (2016); Blei (2016), the objective is to minimize the KL divergence $D_{KL}(P_\theta(w)||P(w|X, Y))$. By applying the evidence lower bound (ELBO) theorem, we can get the equivalent objective to minimize $-\int P_\theta(w) \log P(Y|w, X)dw + D_{KL}(P_\theta(w)||P(w))$. Note that these two terms correspond to likelihood (denoted as $\Delta$) and prior (denoted as $\Gamma$) respectively. Here a prior that all weights follow a 50%-50% Bernoulli distribution is imposed, thus the prior term $\Gamma$ is actually an entropy maximization term (which is widely used in policy gradient to encourage exploration). We refer readers to Appendix A for a more detailed exposition for $\Gamma$. In the main text, we focus on the non-trivial part $\Delta$. Specifically, a training scheme is shown in the second to fifth panel of Fig 1. A complete forward-backward propagation iteration for optimizing $\Delta$ is described as follows:

- In the first forward phase, we sample a set of concrete weights $\overline{w}$ from the distribution $P(w|\theta)$. It is worth mentioning that these concrete weights $\overline{w}$ only exist temporarily.

- In the second forward phase, we do a standard forward propagation on a batch of training data $X$ using concrete weights $\overline{w}$. Denoting the network output as $Y^*$ and the cross entropy metric as $\Delta(*, *)$, we can evaluate $\Delta(Y^*, Y)$.

- In the first backward phase, we do a standard backward propagation, getting gradients w.r.t. concrete weights $\overline{w}$. We denote these gradients as $\frac{\partial \Delta}{\partial \overline{w}}$.

- In the second backward phase, we update the policy network parameters $\theta$. However, since the derivate $\frac{\partial \overline{w}}{\partial \theta}$ cannot be trivially evaluated, we propose to update $\theta$ using the REINFORCE algorithm J William (1992), treating $\mu(\frac{\partial \Delta}{\partial \overline{w}})$ as a pseudo reward. The function $\mu(*)$ will be elaborated later. Once $\theta$ updated, concrete weights $\overline{w}$ are discarded.

The network weights $\overline{w}$ used for the target task $f : X \to Y$ are sampled from $P(w|\theta)$ at the beginning of every single iteration and discarded at the end of the iteration. The de facto parameters we maintain during the whole training procedure are $\theta$, which is a parameterization of $P(w|X, Y)$. Intuitively, we call this burn-after-reading style training scheme as SnapQuant. Note that the regularization term $\Gamma$ is only involved in the second backward phase. Then we describe these four phases sequentially, for which we elaborate the nested parameter structure of the policy network as a necessary foundation.

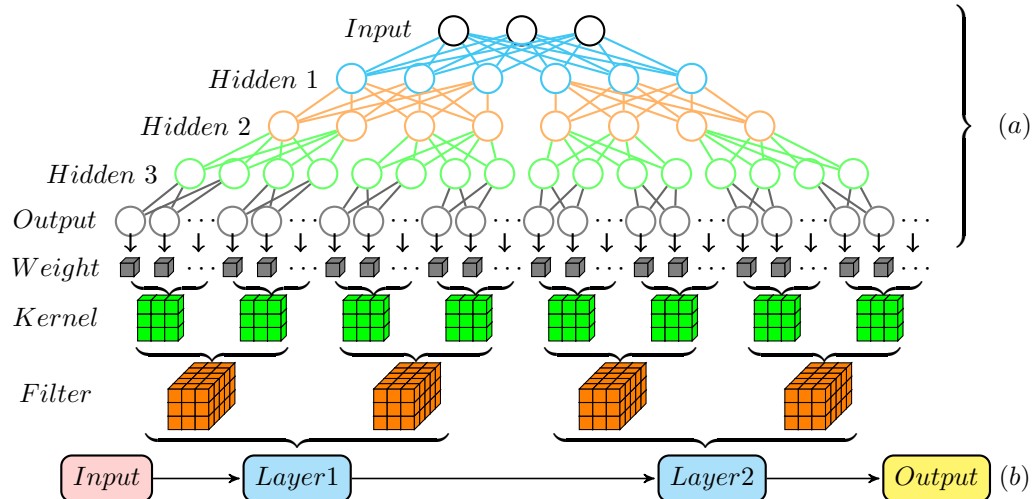

Figure 2: An illustration of the nested parameter structure. The policy network (a) has four layers while the recognition network (b) is a 2-layer CNN. Input to the policy network is state $s$. Blue/orange/green/black connections are **layer shared/filter shared/kernel shared/weight specific** parameters of the policy network. The details of sampling the recognition network weights $\overline{w}$ from the policy network can be found in the main text.

## 3.2 NESTED PARAMETER STRUCTURE

Firstly, we formally define the recognition network. For a convolutional layer indexed by $l$ ($l \in [1, L]$), the weight set is a $O \times I \times K \times K$ tensor, where $O/I$ are output/input channel numbers while $K$ is the spatial size. As illustrated by Fig 2, formally we call every group of $K \times K$ weights *a kernel*, every group of $I \times K \times K$ weights *a filter* and every group of $O \times I \times K \times K$ weights *a layer*. By treating all weights in a kernel as a single dimension, we index a certain weight as $w_{liok}$, $l \in [1, L], i = \in [1, I], o \in [1, O], k \in [1, K^2]$.

There are various ways to construct the policy network. For example, the most straightforward choice is to condition every $w_{liok}$ on an independent $\theta_{liok}$. Adopting this formulation is similar to the probabilistic version of BinaryConnect Courbariaux et al. (2015) except that in SnapQuant the training is done with the REINFORCE algorithm J William (1992). However, this trivial formulation fails to capture the joint distribution of different weights, which is obviously not a realistic assumption. To this end, we propose a **nested parameter structure** of $\theta$, hierarchically modeling the dependency between weights across layers, filters and kernels.

We illustrate our nested parameter structure in Fig 2, where the policy network is a four-layer neural network. Note that these four layers are not conventional fully connected layers as the input-output connections are separated into groups. Now we detail how these groups are organized. Let's denote the input to the policy network as $s$, and the hidden units of subsequent three layers as $h^1$, $h^2$ and $h^3$. Then we name the layer-wise parameters of the policy network according to the different colors used in Fig 2. The blue connections are **layer shared** parameters $\theta_l^1$. The orange connections are **filter shared** parameters $\theta_{li}^2$. The green connections are **kernel shared** parameters $\theta_{lio}^3$. The black connections are **weight specific** parameters $\theta_{liok}^4$.

For a clear exposition of the nested structure, here we formally describe the forward procedure of the policy network. We use $f(*; \theta)$ to represent two sequent operations: (1) feeding inputs/hidden units to these grouply-separated linear layers; (2) feeding outputs of (1) to a subsequent sigmoid nonlinear function that gives $f(*; \theta)$ a probabilistic interpretation. Meanwhile we denote the final outputs of the policy network as $p_{liok}$.

The following equation represents calculating **layer shared** hidden units $h^1$:

$$h_l^1 = f(s; \theta_l^1) \qquad (1)$$

Similarly, **filter shared/kernel shared** hidden units $h^2/h^3$ and final outputs $p_{liok}$ are calculated by:

$$
\begin{aligned}
h_{li}^2 &= f(h_l^1; \theta_{li}^2) \\
h_{lio}^3 &= f(h_{li}^2; \theta_{lio}^3) \\
p_{liok} &= f(h_{lio}^3; \theta_{liok}^4)
\end{aligned}
\tag{2}
$$

As such, we get the **weight specific** outputs $p_{liok}$. Each $p_{liok}$ characterizes a policy:

$$
\begin{cases}
P(\overline{w}_{liok} = +1) = p_{liok} \\
P(\overline{w}_{liok} = -1) = 1 - p_{liok}
\end{cases}
\tag{3}
$$

Following these policies, we can do the sampling mentioned in the first forward phase. Thanks to the nested parameter structure, the policies to generate different weights are connected by those shared weights. For clarity, we further give a concrete example to demonstrate the parameter sharing mechanism. Imagine there are two weights $w_{liok_1}$ and $w_{liok_2}$ which reside in the same kernel. They are sampled according to $p_{liok_1}$ and $p_{liok_2}$. Formally these two quantities are calculated by:

$$
\begin{aligned}
p_{liok_1} &= f(f(f(f(s; \theta_l^1); \theta_{li}^2); \theta_{lio}^3); \theta_{liok_1}^4) \\
p_{liok_2} &= f(f(f(f(s; \theta_l^1); \theta_{li}^2); \theta_{lio}^3); \theta_{liok_2}^4)
\end{aligned}
$$

As clearly shown by these two equations, among all quantities used to generated $p_{liok_1}$ and $p_{liok_2}$, the only difference is the **weight specific** parameters $\theta_{liok_1}^4$ and $\theta_{liok_2}^4$.

Now that the first forward phase has been formally defined, we move on to the remaining three phases. Recall that the second forward phase and the first backward phase actually form a ordinary forward-backward routine of a typical CNN. Thus the only unclear part is the second backward phase, which will be elaborated in the next subsection.

### 3.3 OPTIMIZING THE EXPECTED LIKELIHOOD WITH REINFORCE

Since the procedure of sampling $\overline{w}$ from $P(w|\theta)$ is not trivially differentiable, we propose to update the parameters $\theta$ of the policy network using the REINFORCE algorithm J William (1992).

We associate a total of $L \times I \times O \times K^2$ Markov Decision Processes (MDPs) with all the weights $\overline{w}$. All these MDPs share the same state $s$, which is the globally shared input vector into the policy network. For every policy characterized by $p_{liok}$, there are two actions: (1) assigning value +1 to $\overline{w}_{liok}$ (with probability $p_{liok}$); (2) assigning value -1 to $\overline{w}_{liok}$ (with probability 1-$p_{liok}$). To this end, we have to define rewards w.r.t. these two actions.

Here, we propose an intuitive trick that assumes there is only one step in an episode of each MDP and an episode corresponds to a complete forward-backward iteration in Fig 1. As such, we can define pseudo rewards $r = \mu(\frac{\partial \Delta}{\partial \overline{w}})$, which is related to the gradients w.r.t. $\overline{w}$ obtained in the current iteration.

In our implementation, the pseudo reward function $\mu(*)$ takes this form:

$$
\begin{aligned}
g_{liok} &= \frac{\partial \Delta}{\partial \overline{w}_{liok}} \\
r_{liok} &= \mu(g_{liok}) = -\beta \times g_{liok} \times \overline{w}_{liok}
\end{aligned}
\tag{4}
$$

Note that $\overline{w}_{liok}$ is either $+1$ or $-1$ so multiplying $\overline{w}_{liok}$ only changes the sign of the reward and the magnitude of the reward equals $\|\beta g_{liok}\|$. $\beta$ is the scaling factor for the reward, which is a hyper-parameter. The reward gives a positive value if the gradient $g_{liok}$ and the sampled weight $\overline{w}_{liok}$ take different signs. For intuitively understanding why we design rewards as such, we give a concrete example. Assume $\overline{w}_{liok}$ is sampled as $+1$ during the second forward phase and the gradient on it $g_{liok}$ emerges as negative. According to the gradient descent rule, we should update $\overline{w}_{liok}$ into $\overline{w}_{liok} - g_{liok}$. Obviously, $\overline{w}_{liok} - g_{liok} > \overline{w}_{liok}$, meaning that if this is not a binary network, $\overline{w}_{liok}$ would take a value larger than $+1$ (e.g. $+2$) after this update. As $\overline{w}_{liok}$ can only take the value $+1$ or

$-1$, instead we increase the possibility of assigning $+1$ to $\overline{w}_{liok}$. In other words, we increase $p_{liok}$, meaning that the action of assigning $+1$ to $\overline{w}_{liok}$ is encouraged. In this case, the reward is positive.

Now that the states, actions and rewards of these 1-step MDPs have already been defined, we can update $\theta$ using the standard REINFORCE algorithm J William (1992). The expected reward is:

$$J(\theta) = \sum_{l,i,o,k} E_{P(w_{liok}|\theta)}(r_{liok})$$

According to the REINFORCE principle, we have:

$$\bigtriangledown J(\theta) = \sum_{l,i,o,k} E_{P(w_{liok}|\theta)}(\bigtriangledown_\theta \log P(w_{liok}|s;\theta) \times r_{liok})$$

Via applying a Monte Carlo sampling, we get an unbiased estimator of this quantity:

$$\bigtriangledown J(\theta) = \sum_{l,i,o,k} (\bigtriangledown_\theta \log P(\overline{w}_{liok}|s,\theta) \times r_{liok}) \tag{5}$$

Now that we have $\bigtriangledown J(\theta)$ which optimizes the first term in the aforementioned objective $\Delta + \Gamma$, combining it with $\bigtriangledown \Gamma(\theta)$ (for details see Appendix A) drives the variational distribution $P_\theta(w)$ to the minimum of $D_{KL}(P_\theta(w)||P(w|X,Y))$. The whole training procedure of our SnapQuant is summarized as in Algorithm 1.

---

**Algorithm 1** SnapQuant for training BWNs from scratch

---

**Require:** $X$: natural images, $Y$: image labels, $w$: recognition network weights restricted to $+1$ or $-1$ (what we really need is $w$'s structure), $Y^*$: recognition network outputs, $\Delta(*,*)$: cross entropy function, $\{\theta: \theta^1, \theta^2, \theta^3, \theta^4\}$: randomly initialized policy network parameters, $s$: the globally shared state vector to the policy network, $N$: maximum iteration number, $\beta$: pseudo reward scaling factor.
**Ensure:** $P(w|\theta)$ approximates the posterior distribution $P(w|X,Y)$.
 1: **for** $n = 1, 2, \ldots, N$ **do**
 2:     Calculate $\{p_{liok}\}$ according to Equation (1-2), using $s$ and $\theta$.
 3:     Sample concrete weights $\overline{w} = \{\overline{w}_{liok}\}$ according to Equation (3).
 4:     Sample a mini-batch $B$ from the training set $\{X,Y\}$.
 5:     Do a standard feed forward using concrete weights $\overline{w}$ and $B$, getting $\Delta(Y^*,Y)$.
 6:     Do a standard backward propagation to get gradients $g_{liok} = \frac{\partial \Delta}{\partial \overline{w}_{liok}}$.
 7:     Calculate pseudo rewards according to Equation (4).
 8:     Update $\theta$ according to $\bigtriangledown J(\theta)$ in Equation (5) and $\bigtriangledown \Gamma(\theta)$ in Appendix A.
 9:     Discard concrete weights $\overline{w}$.
10: **end for**

---

## 4 EXPERIMENTS

### 4.1 EXPERIMENTAL SETTINGS

We evaluate SnapQuant with standard MNIST Yann et al. (1998), CIFAR-10/100 Krizhevsky & Hinton (2009) and ImageNet Russakovsky et al. (2015) datasets. At the inference phase, we use the trained policy network to sample concrete binary weights $\overline{w}$ for evaluation. Assuming the recognition network has $|\overline{w}|$ weights, the amount of possible binary networks is up bounded by $2^{|\overline{w}|}$, which can be considered as infinite. For easy implementation, we sample 100 binary networks from the policy network and pick the best one. This inference scheme demonstrates the versatility of SnapQuant's probabilistic nature. In the experiments, four different settings (namely non-sharing, kernel-wise sharing, filter-wise sharing and layer-wise sharing) are considered for SnapQuant, according to how parameters are shared in the nested policy network. For instance, SnapQuant (layer-wise sharing) means the complete nested parameter structure is exploited, while SnapQuant (non-sharing) represents the trivial solution of conditioning every $w_{liok}$ on an independent $\theta_{liok}$. Taking 32-bit

floating-point network model as the baseline, we compare our method with the most related method BinaryConnect Courbariaux et al. (2015). BinaryConnect has two versions: (1) the deterministic version binaries weights using a hard threshold; (2) the stochastic version also samples concrete weights at the inference phase like our method, thus we also report its best result out of 100 trials. All algorithms are implemented with Pytorch Paszke et al. (2017).

## 4.2 Result Comparison on MNIST

MNIST is a digit recognition dataset consisting of 60000 training samples and 10000 testing samples. We adopt the LeNet-5 architecture Yann et al. (1998) and the results are summarized in Table 1. It can be seen that SnapQuant reaches an error rate of $0.89\%$, which is better than the stochastic version of BinaryConnect, and is very close to the performance of the full-precision baseline.

| Architecture | Method | Error rate (%) |
|---|---|---|
| LeNet-5 | 32-bit floating-point baseline | 0.81 |
| | BinaryConnect (deterministic) | 0.88 |
| | BinaryConnect (stochastic) | 1.24 |
| | SnapQuant (layer-wise sharing) | 0.89 |

Table 1: Quantitative results on MNIST.

## 4.3 Result Comparison on CIFAR-10/100

CIFAR-10 Krizhevsky & Hinton (2009) consists of 50000 training images and 10000 testing images collected from 10 classes. CIFAR-100 has the same numbers of training/testing images containing 100 classes. For CIFAR-10, we firstly evaluate the VGG-like architecture proposed in BinaryConnect Courbariaux et al. (2015). It has 14.03 million parameters, causing severe over-fitting. The 32-bit floating-point baseline can only reach $10.64\%$ error rate, which is worse than SnapQuant and BinaryConnect as binarization can be considered a regularization. Here, for BinaryConnect, we use its best result reported in the original paper. Then we evaluate the ResNet-20 architecture He et al. (2016) on both CIFAR-10 and CIFAR-100. The results are shown in Table 2. On both benchmarks, SnapQuant outperforms the stochastic version of BinaryConnect by large margins while performs comparably with its deterministic version. It is worth mentioning that the deterministic version cannot provide the versatility of Bayesian formulation, such as model selection and uncertainty measure.

| Architecture | Method | Error rate on CIFAR-10 (%) | Error rate on CIFAR-100 (%) |
|---|---|---|---|
| VGG-like | 32-bit floating-point baseline | 10.64 | |
| (14.03M) | BinaryConnect (deterministic) | 9.90 | |
| | BinaryConnect (stochastic) | 8.27 | |
| | SnapQuant (layer-wise sharing) | 9.26 | |
| ResNet-20 | 32-bit floating-point baseline | 8.75 | 31.20 |
| (0.27M) | BinaryConnect (deterministic) | 10.85 | 41.99 |
| | BinaryConnect (stochastic) | 17.22 | 55.98 |
| | SnapQuant (non-sharing) | 14.24 | 46.74 |
| | SnapQuant (kernel-wise sharing) | 13.39 | 44.43 |
| | SnapQuant (filter-wise sharing) | **12.74** | 43.46 |
| | SnapQuant (layer-wise sharing) | 13.29 | **43.13** |

Table 2: Quantitative results on CIFAR-10/100.

## 4.4 Result Comparison on ImageNet

ImageNet classification dataset is known as the most challenging image classification benchmark so far. It has about 1.2 million training images and 50 thousand validation images. Each image is annotated as one of 1000 object classes. We apply our SnapQuant to two popular CNN architectures: AlexNet Krizhevsky et al. (2012) and ResNet-18 He et al. (2016). Using the center crops of validation images, we report the results with two standard measures: top-1 error rate and top-5 error rate. The results are summarized in Table 3. Results for comparison methods are taken from the original papers Rastegari et al. (2016). On AlexNet, SnapQuant outperforms BinaryConnect by $16.9\%$ and is worse than Binary-weight-network by $4.5\%$, considering top-1 error rate. On ResNet-18,

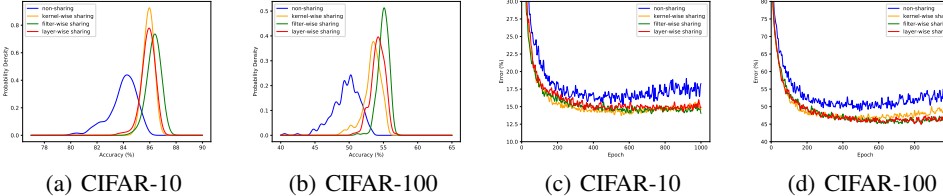

| (a) CIFAR-10 | (b) CIFAR-100 | (c) CIFAR-10 | (d) CIFAR-100 |

Figure 3: (a/b) The accuracy distribution of 100 randomly sampled binary networks for different SnapQuant settings, on CIFAR-10/100. (c/d) Testing set error curves of different SnapQuant settings on CIFAR-10/100. Each data point corresponds to the performance of a randomly sampled network at that epoch. Note these four figures are better viewed electronically for a full resolution.

SnapQuant's top-1 performance gap against Binary-weight-network is 1.9%. As a reminder, Binary-weight-network exploits filter-wise scaling factors leading to $+\alpha_i$ or $-\alpha_i$ weights. Scaling involves multiplication operations, which costs additional hardware cycles. BinaryNet and XNOR-Net exploit 1-bit activations. Recall that the central scientific problem considered in this paper is to learn the posterior distribution of binary weights, learning the distribution of binary activations is another interesting direction yet out of our scope.

Table 3: Error rates (%) on ImageNet validation set.

| Method | AlexNet | | ResNet-18 | |
| --- | --- | --- | --- | --- |
| | Top-1 | Top-5 | Top-1 | Top-5 |
| 32-bit floating-point baseline | 43.4 | 19.8 | 30.7 | 10.8 |
| BinaryNet (1-bit activation) | 72.1 | 49.6 | — | — |
| XNOR-Net (1-bit activation) | 55.8 | 30.8 | 48.8 | 26.8 |
| Binary-weight-network (w/ scaling) | 43.2 | 20.6 | 39.2 | 17.0 |
| BinaryConnect (deterministic) | 64.6 | 39.0 | — | — |
| SnapQuant (layer-wise sharing) | 47.7 | 25.0 | 41.1 | 18.3 |

## 4.5 NESTED PARAMETER STRUCTURE: DEEP ANALYSIS

For a deep understanding of the nested parameter structure, we present ablation studies on CIFAR-10/100. According to the results shown in Table 2, all parameter sharing settings clearly outperform the non-sharing setting. For CIFAR-10, the filter-wise sharing setting gets an error rate of $12.74\%$, which outperforms the non-sharing baseline by $1.50\%$. For CIFAR-100, the layer-wise sharing setting performs best, surpassing the non-sharing baseline by $3.61\%$. A deeper analysis is presented in Figure 3-a/b, in which the accuracy distribution of 100 randomly sampled binary networks is illustrated. Note that the accuracy variances of the sharing settings are obviously smaller than the non-sharing baseline. According to the mean accuracies of the different settings, we can see that the filter-wise sharing performs the best. This implies that modeling the joint distribution across layers is not that necessary, which is consistent to common senses. While kernels or filters may be statistically related, weights in different layers are less likely to correlated with each other. We further present the testing error curves in Figure 3-c/d, showing that the sharing settings enable a more stable training.

Variance reduction results and training details are provided in Appendix B and C.

## 5 CONCLUSIONS

In this paper, we proposed SnapQuant, a probabilistic method for training binary weight neural networks from scratch under the Bayesian deep learning perspective. We approximate the posterior distribution of binary weights with a reinforcement learning scheme. A policy network with a novel nested parameter structure was presented to parameterize the posterior distribution of binary weights. We show that the proposed method performs well in several visual recognition tasks including ImageNet, as tested with different network architectures.

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

## A  OPTIMIZING THE REGULARIZATION TERM $\Gamma$

Recall that the variational approximation objective is $\Delta + \Gamma$, in which:

$$\Delta = - \int P_\theta(w) \log P(Y|w, X) dw$$

$$\Gamma = D_{KL}(P_\theta(w)||P(w))$$

While the main paper focuses on optimizing $\Delta$ w.r.t. $\theta$ using $\bigtriangledown J(\theta)$, here we show how to optimize the second term $\Gamma$. Note that it is imposed on the policy network thus only involved in the second backward phase in Fig 1. Recall that (see Equation 3) for a certain concrete $\overline{w}_{liok}$:

$$P_\theta(\overline{w}_{liok}) = \left\{ \begin{array}{l} P(\overline{w}_{liok} = +1) = p_{liok} \\ P(\overline{w}_{liok} = -1) = 1 - p_{liok} \end{array} \right. \tag{6}$$

We adopt the 50%-50% Bernoulli distribution as the prior:

$$P(\overline{w}_{liok}) = \left\{ \begin{array}{l} P(\overline{w}_{liok} = +1) = 0.5 \\ P(\overline{w}_{liok} = -1) = 0.5 \end{array} \right. \tag{7}$$

Thus the second term $\Gamma$ is evaluated as:

$$\Gamma = \sum_{l,i,o,k} D_{KL}(P_\theta(\overline{w}_{liok})||P(\overline{w}_{liok}))$$

Each quantity in the summation can be calculated as:

$$p_{liok} \log \frac{p_{liok}}{0.5} + (1 - p_{liok}) \log \frac{1 - p_{liok}}{0.5}$$

$$= p_{liok} \log p_{liok} + (1 - p_{liok}) \log(1 - p_{liok}) - \log 0.5$$

$-\log 0.5$ is a constant. $p_{liok} \log p_{liok} + (1-p_{liok}) \log(1-p_{liok})$ is the negative entropy so minimizing it means maximizing the entropy. Note that entropy maximization is a widely used technique in REINFORCE to encourage exploration. Interestingly, it functions as the regularization term which shapes the variational approximation $P_\theta(w)$, in the formulation of SnapQuant.

More specifically, $\bigtriangledown \Gamma(\theta) = \frac{\partial \Gamma}{\partial p} \frac{\partial p}{\partial \theta}$. $\frac{\partial \Gamma}{\partial p}$ is the derivative of aforementioned negative entropy while $\frac{\partial p}{\partial \theta}$ follows the standard back propagation of the policy network, which consists of fully connected layers, slicing layers and sigmoid activation layers.

## B  VARIANCE REDUCTION FOR REINFORCE

An experiment on CIFAR-10 using ResNet-20 is given in Fig 4. We incorporate a running mean baseline to reduce the variance of gradients. We can see that using a baseline speeds up convergence (see training curves before 100 epochs) yet cannot stably improve the performance on validation set.

Figure 4: Using a running mean baseline for variance reduction on CIFAR-10.

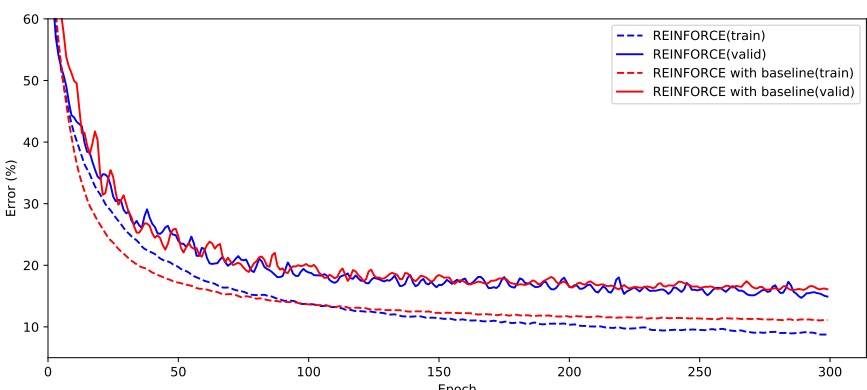

## C  NETWORK TRAINING DETAILS

**LeNet-5** For the MNIST dataset, we use the LeNet-5 architecture. For all the experiments in Table 1, we use Adam optimizer with the initial learning rate 0.01 and an exponential learning rate decay strategy at the end of each epoch, where the decay rate is 0.9. We set batch size to 64 with BN to speed up the training. The training is run for 100 epochs. Noted that in our SnapQuant experiments, we use another Adam optimizer to train our policy network, with the initial learning rate 0.1 and an exponential learning rate decay strategy at the end of each epoch, where the decay rate is 0.9. The reward scaling factor $\beta$ is 0.1.

**VGG-like** We follow the CNN architecture in the BinaryConnect Courbariaux et al. (2015), implement the same VGG-like network expect that we replace the finial SVM by a softmax classifier, and we do not quantize this classifier. For all the experiments of this VGG-like architecture in Table 2, we use Adam optimizer with the initial learning rate 0.1 and an exponential learning rate decay strategy at the end of every 50 epochs, where the decay rate is 0.5. We set batch size to 128 with BN to speed up the training and the training is run for 300 epochs. Noted that in our SnapQuant experiments, we use another Adam optimizer to train our policy network, with the initial learning rate 0.01 and an exponential learning rate decay strategy at the end of every 10 epochs, where the decay rate is 0.9. The reward scaling factor $\beta$ is 0.1.

**ResNet-20** We implement the ResNet-20 topological structure and do not quantize the first convolutional layer and the last classifier like most of other methods. For all the experiments of ResNet-20 architecture in Table 2, we use Adam optimizer with the initial learning rate 0.01 and an exponential learning rate decay strategy at the end of every 50 epochs, where the decay rate is 0.5. We set batch size to 256 with BN to speed up the training and the training is run for 1000 epochs. Noted that in our SnapQuant experiments, we use another Adam optimizer to train our policy network, with the initial learning rate 0.01 and an exponential learning rate decay strategy at the end of every 10 epochs, where the decay rate is 0.9. The reward scaling factor $\beta$ is 0.1.

**AlexNet** Experiments with AlexNet are conducted on a server with 4 Titan X GPUs (while we use 1 GPU for experiments with the aforementioned three networks), and we follow standard experimental settings Rastegari et al. (2016). To train binary AlexNet, we run SnapQuant with Adam optimizer for 100 epochs with the batch size of 256, the weights decay of 0.0001 and the momentum of 0.9. The learning rate starts at 0.01 and is divided by 10 every 30 epochs. Noted that in our SnapQuant experiments, we use another Adam optimizer to train our policy network, with the initial learning rate 0.01 and a learning rate decay strategy at the end of every 10 epochs, where the decay rate is 0.0002. The reward scaling factor $\beta$ is 0.1.

**ResNet-18** Experiments with ResNet-18 are conducted on a server with 4 Titan X GPUs, and we follow standard experimental settings Rastegari et al. (2016). To train binary ResNet-18, we run SnapQuant with Adam optimizer for 100 epochs with the batch size of 256, the weights decay of 0.0001 and the momentum of 0.9. The learning rate starts at 0.01 and is divided by 10 every 30

epochs. Noted that in our SnapQuant experiments, we use another Adam optimizer to train our policy network, with the initial learning rate 0.01 and a learning rate decay strategy at the end of every 10 epochs, where the decay rate is 0.0002. The reward scaling factor $\beta$ is 0.1.

