# OpenReview forum: "SnapQuant: A Probabilistic and Nested Parameterization for Binary Networks"
_ICLR.cc/2019/Conference_

### Official Review · AnonReviewer2 · 2018-10-31
**learning binary weight neural networks using a structured variational approximation, gradients estimated using modified reinforce**

**Rating:** 5
**Confidence:** 4

**Review:**

Summary: The paper considers a variational inference strategy for learning neural networks with binary weights. In particular, the paper proposes using a structured recognition model to parameterise the variational distribution, which couples the weights in different layers/filters in a non-trivial way. The gradient of the expected likelihood term in the variational lower bound is estimated using the REINFORCE estimator. This paper adjusts this estimator to use the gradient of the log-likelihood wrt the samples. Experiments on several image classification tasks are provided.

evaluation:

pros:
- the idea of the proposed approach is interesting: using variational inference for binary weight neural networks. While recent work on VI for discrete variables only focused on discrete latent variable models, this work shows how VI can be used for binary neural networks.

cons:
- the writing, in my opinion, needs to be improved [see my comments below]. The VI presentation is cluttered and the justification of using the pseudo-reward for reinforce is not clear.
- the experimental results are mixed and it's not clear to me how to interpret them/compare to the baselines -- what is the goal here: computational efficiency, compression or accuracy?

Some specific questions/comments:

+ What is the input of the policy/recognition network? It's not clear from the paper whether this includes the inputs of the current batch or outputs or both? If so, how are variable batch sizes handled? What is the input to this network at test time? In contrast to generative models/VAEs, the weights here are global parameters and it's not clear to me these should be varied for different data batches.

+ related to the question above: how is prediction handled at test time? Say the parameters of the variational distribution over weights are generated using the recognition network, then 100 weights are sampled given these parameters which then give 100 predictions -- should these be then averaged out to get the final prediction? I'm not quite sure I understand why the paper chose to *pick the best one* out of 100 predictions and the justification/criterion for this procedure.

+ The writing is not very clear at places, and it does not help that the references being merged with the text. I'm also not sure about some of the technical jargons/terms used in the papers:
- reinforcement learning: is this really a reinforcement learning problem? If you tackle this problem from a pure variational perspective, reinforce is used to obtain the gradient of the expected log-likelihood wrt the variational parameters. But instead of using the log likelihood, a learning signal that depends on the gradient of the log-likelihood is used.
- concrete weights -- what are these? I assume they are just binary weights sampled from the variational approximation.
- middle of page 3: p(w|X, Y) = p_\theta(w): this is not precise as p_\theta(w) is only an approximation to the exact posterior, which then allows us to lower bound the log marginal likelihood. "common practice in modern variational approximation": This is the standard way of deriving the lower bound and has been used for many years.

+ the reinforce estimator tends to have high variances since it does not make use of the gradient of the function in the expectation. This paper adjusts the vanilla estimator with a learning signal that involves the gradient. Could you comment on the bias/variance trade-off of the resulting estimator? Much of recent literature on learning discrete variables, as far as I understand, propose ways to not to have to use the vanilla reinforce, for example Concrete, Relax or rebar, albeit the focus on latent variable models.

+ model selection and uncertainty measure: the paper mentions these potential advantages of the proposed approach over deterministic binarisation schemes, but does not fully explore and test these.

---

> ### Author Response · Authors · 2018-11-27
> **Thanks for the attentive reading and professional comments.**
>
> - We thank R2 for pointing out the issue w.r.t. the pseudo-reward for reinforce. For now, we admit it is ad-hoc. We will try to formulate it in a more elegant manner in a future version.
> - We admit that the proposed method is not better than baselines. A potential advantage is versatility of the formulation. However, as rightly pointed out by reviewers, it is not validated by experiments. We will try to demonstrate the versatility in a future version.
> - Currently, the input to the policy network is a globally shared state vector that remains constant. We thank R2 for these professional suggestions on augmenting it into more complex forms.
> - We admit that the experimental protocol is not very convincing in this manuscript. We will use the mean values and variances in a future version.
> - We thank R2 for pointing out the issue about the gradient estimator. We will try to give a better formulation in a future version.
> - Yes, concrete weights refer to binary weights sampled from the variational approximation.
> - We thank R2 for this professional suggestion. We will remove 'modern' in a future version.
> - Actually, I have to admit that I don't know how to answer this question. What is concrete, relax and rebar? As for the bias-variance trade-off, we will try to outline its envelope by varying the prior from 0.1-0.9 to 0.9-0.1, in a future version.
> - We will incorporate experiments showing these features in a future version.

---

### Official Review · AnonReviewer3 · 2018-11-01
**Need clarification on the experimental setting**

**Rating:** 6
**Confidence:** 3

**Review:**

This paper proposes to binarize all parameters of a CNN where the binary parameters are generated from another policy neural network (let's call it parameter generator). The parameter generator network has a special nested structure to regularize parameters within layers and filters. All parameters in CNN and parameter generator network are jointly trained. Since the gradient is hard to back propagated through binary variables, the paper adopts reinforcement learning approach to back-propagate rewards to the parameter generator.

The experiments look solid. The results show that the proposed approach is slightly worse than BinaryConnect (baseline) on MINST, CIFAR10 and CIFAR100, but outperforms BinaryConnect on ImageNet by a large margin. The ablation study also verified the need for the proposed nested parameter structure.

The paper is well written. The proposed method (1) is able to provide the posterior distribution of parameters so that we can use that in other applications such as confidence estimation and model selection (2) is memory and power efficient due to binarization.

Having said that, the experimental setting falls short. For the proposed method, the paper samples 100 binary networks from the parameter generator and pick the best one, which seems not correct. The results of the proposed method should be based on the average of these 100 binary networks, rather than picking the best one, because we won't be able to know which binary network is the best. Using the best one seems label leakage to me. For now, I'll give the benefit of doubt. Please clarify this during the feedback phase.

---

> ### Author Response · Authors · 2018-11-27
> **Thanks for the attentive reading and professional comments.**
>
> We thank R3 for these encouraging feedbacks on the proposed training scheme and the nested structure.
> We want to clarify our results obtained by our sampled binary networks. For both baseline and our method, we evaluate the model (just sample one binary model at each epoch) on valid-set at the end of each training epoch and report the best one among the results of last 100 training epochs, so it's a fair comparison. We admit that the experimental methodology is not very convincing in the manuscript. We will incorporate a systematic evaluation with mean values and variances, in a future version.

---

### Official Review · AnonReviewer1 · 2018-11-02
**Bayesian view on deep network binarization - interesting idea but lacks in clarity and experiments**

**Rating:** 4
**Confidence:** 5

**Review:**

The paper treats network binarization as learning a bernoulli probability that each weight is 0 or 1. It uses a hierarchical approach to factor the probability in to parameters shared between weights/kernels/layers. The result is a network that can be used to sample a binary network. The main novelty is that full-precision weights are not needed during training.

- There is a variance issue underlying this idea. The goal is to find a binary network that has high performance, but the probabilistic formulation could have a high variance in terms of the performance of the resulting binary networks. In the experiments, the variance is not shown, rather the authors sample 100 networks and pick the best one based on validation set accuracy? (they just say "pick the best one"). Should this be accounted for in the objective function? The prior has very high variance (p=0.5). The authors claim that this "demonstrates the versatility" -- how?
- Related to the above, there is a high variance associated with he REINFORCE estimator. In the appendix, the authors use a baseline value to alleviate this, but no discussion is provided in the main text or the experiments.
- For the stochastic version of binary connect, the authors report the best out of 100 trials. I would rather like to see the mean and confidence intervals, for this as well as the proposed method.

- Please provide a comparison of the number of hyper parameters used vs the number of binary network parameters. Is it feasible to store the "master network" in memory for small devices? It seems you need more parameters than the original network as you have weight specific parameters + kernel/filter/layers specific parameters? Is there any generalization between these hyperparameters that can be shown in experiments e.g. using a compact hierarchy?
- More generally, how do you see this method being used in practice? Do you sample each binary network on the device?
- How is this better than other methods of training binary nets, which have better accuracy than your approach (according to Table 2)?
- In the experiments, the hierarchical structure used for hyperparameters is not clearly described.

- I found the exposition in Section 3.2 to be very confusing using f(*)  whereas it is very simply described in words. The policy network is hierarchical upon layers/filters/kernels/weights. What is "s" in equations (1) etc.?
- In Section 3.3, I found the connection to MDPs tenuous, whereas it is easy to understand that you are using REINFORCE to estimate the gradient of the expectation.
- The pseudo-reward is completely ad-hoc. Since this is a "1-step MDP" (bandit problem?), the reward is maximized when the probability of w=1 is related to the sign of the gradient. In the end, we seem to have arrived at something that is basically similar to BinaryConnect (i.e. using the sign of the gradient).
- Only experiments with an uniform prior are shown. Can the prior be used in some ways?

- In the experiments, the activation functions used are not described.
- Some numbers are missing in Table 3.
- One advantage of TernaryConnect is that it allows sparsity by allowing zero weights, whereas binary nets only allow +/- 1 weights. That is, Ternary Connect should not be dismissed simply and the performance of Ternary Connect should be shown on all the datasets.
- The error for VGG on CIFAR-10 is very high compared to SOTA (close to 94% accuracy).
- The legend on Figure 3 is not readable, the font size can be increased.
- Page 1 "regardless of the availability ..." - citation needed.
- Page 2, experiments are not a separate contribution.

---

> ### Author Response · Authors · 2018-11-27
> **Thanks for the attentive reading and professional comments.**
>
> We thank R1 for the attentive reading and professional comments. We admit that our experiments are insufficient and will improve them in a future version.
>
> - We thank R1 for pointing out the variance issue and the lack of experiments w.r.t. versatility. Fig.3-a/b is related to the variance issue. We will include a systematic variance evaluation into a future version. We will provide experiments w.r.t. model selection and uncertainty estimation to demonstrate the versatility.
> - We will give a more detailed discussion w.r.t. the variance of the gradient estimator, in a future version.
> - We will improve the experimental mythology in a future version, reporting mean values and confidence intervals.
>
> - Yes, the master network is bigger than the original network by 2~4 times of model size, but the master network is not designed for running on small devices, because what we actually need is the sampled binary network, which is efficient for small devices. We thank R1 for pointing out the possibility of exploring the generalization of the master network.
> - We assume the scenario in which we sample a network on a small validation set to address domain drift issues, then deploy sampled weights onto the device. Unfortunately, as rightfully pointed out by R1, this versatility is not shown in this manuscript. We will incorporate experiments showing this versatility in a future version.
> - As mentioned in the last bullet point, we expect the master network to give a better accuracy on a slightly different CIFAR, after sampling. However, since this is not validated in this manuscript, we admit it is just a potential advantage.
> - We will describe it more clearly in a future version.
>
> - Yes, the policy network is hierarchical upon layers/filters/kernels/weights and we design it for two purposes: a) make the distribution of binary weights related to each other based on their position in CNN architecture. b) reduce the model size of the policy network. We thank R1 for suggesting expressing f(*) with formulas. s is the globally shared state vector to the policy network.
> - We thank R1 for pointing out the issues w.r. MDPs. We will try to improve it.
> - We admit that our method is inspired by the stochastic version of BinaryConnect. If we do not use the nested structure and do not share the parameters of policy, our method will degenerate to a no-share one which is similar to BinaryConnect. Nevertheless, we find the stochastic version of BinaryConnect is hard to train, so we design the nested structure to improve it and use reinforce algorithm to bridge the output of the policy network and sampled binary weight as the sampling process is not differentiable. We admit the pseudo-reward is ad-hoc. We will try to dig deep into the design principle of this reward.
> - We will include experiments using other priors, in a future version.
>
> - We will give a more detailed description of the architecture.
> - We will include these missing numbers in a future version.
> - We will incorporate comparisons with Ternary Connect.
> - We will try to achieve a higher accuracy, perhaps by improving the implementation.
> - We will improve the figures in a future version
> - We will add citations after 'regardless of the availability'
> - We thank R1 for pointing out that experiments are not a separate contribution. We will rephrase this line in a future version.

---

### Meta-Review · Area_Chair1 · 2018-12-15

**Confidence:** 3
**Recommendation:** Reject

**Metareview:**

Reviewers mostly recommended to reject. Please take reviewers' comments into consideration to improve your submission should you decide to resubmit.